# In Vivo Accuracy of a New Digital Planning System in Terms of Jaw Relation, Extent of Surgical Movements and the Hierarchy of Stability in Orthognathic Surgery

**DOI:** 10.3390/jpm12050843

**Published:** 2022-05-21

**Authors:** Thomas Stamm, Eugenia Andriyuk, Johannes Kleinheinz, Susanne Jung, Dieter Dirksen, Claudius Middelberg

**Affiliations:** 1Department of Orthodontics, University of Münster, Albert-Schweitzer-Campus 1, 48149 Münster, Germany; thomas.stamm@ukmuenster.de (T.S.); claudius.middelberg@ukmuenster.de (C.M.); 2Department of Cranio-Maxillofacial Surgery, University Hospital Münster, Albert-Schweitzer-Campus 1, 48149 Münster, Germany; johannes.kleinheinz@ukmuenster.de (J.K.); susanne.jung@ukmuenster.de (S.J.); 3Department of Prosthetic Dentistry and Biomaterials, University Hospital Münster, Albert-Schweitzer-Campus 1, 48149 Münster, Germany; dieter.dirksen@ukmuenster.de

**Keywords:** orthognathic surgery, virtual planning, orthodontics

## Abstract

This retrospective cohort study compares the virtual planned and postoperative jaw positions in patients undergoing orthognathic surgery. Surgery was virtually planned with the Digital Münster Model Surgery system (DMMS). Primary outcome: Spatial difference in the maxillo-mandibulo relation between virtual planning and postoperative result. Secondary outcome: Possible relationship between the measured differences and surgical movements as well as the postoperative stability according to Proffit. Ninety female and sixty-one male patients were included in the study. The average translation errors were 0.54 ± 0.50 mm (anteroposterior), 0.37 ± 0.33 mm (mediolateral), and 0.33 ± 0.28 mm (superoinferior). Orientation errors were 0.86 ± 0.79 degrees (yaw), 0.54 ± 0.48 degrees (roll), and 0.90 ± 0.72 degrees (pitch). The surgical procedures do not differ with respect to their error sizes. Maxilla forward and class II maxilla up with mandible forward are the most precise procedures. Most significant differences were found in the anteroposterior direction, whereby the extent of the surgical movement has no effect on the magnitude of the error. The process of planning with the DMMS followed by surgery is highly accurate and shows error values well below the clinically accepted limit of two millimeters in translation and four degrees in rotation.

## 1. Introduction

The orthodontist and the general dentist play a central role in the patient’s decision making for orthognathic surgery. As the first professional contact person, they have a critical impact on the patient’s awareness of their underlying dentofacial problems and treatment options [1].

Orthognathic surgery treatment is a so-called “Two-Captains-Doctrine” where the surgeon and orthodontist have different responsibilities in their own therapy area [2]. With pre- and/or postoperative orthodontics, the orthodontist is involved for the longest period of treatment and is, in the end, responsible for proper occlusion and masticatory function. Thus, the main orthodontic interest is in a postoperative maxillo-mandibulo arch position that can be treated into an optimal occlusion. From an orthodontic point of view, the accuracy that takes into account all steps up to the stage of postoperative orthodontics is of primary interest.

With the general availability of cone-beam computed tomography (CBCT) in dentistry, the focus of planning has shifted from occlusion to the simulation of osteotomies and jaw positioning, and a variety of assessments has been used to demonstrate the accuracy of these methods. Recent systematic reviews revealed that the literature lacks consensus regarding the validation of 3D planning in orthognathic surgery [3,4]. The focus on the positional relationship of the mandible to the maxilla is also not well represented in studies. It could be shown that planned movements of the individual jaws can be very accurate, but an incorrect condylar position can worsen the interocclusal relation. Validation information should be therefore based on data with a distinct time interval to surgery and without any interocclusal fixation.

Despite the heterogeneity of the available studies, all authors agree on one point, which is that the clinical error size should not exceed 2 mm [5,6,7,8,9]. Another common feature is that the differences in jaw positions are increasingly analyzed in terms of translation and rotation of the three spatial axes, with rotations described as yaw (superoinferior axis), roll (anteroposterior axis), and pitch (mediolateral axis). This characterization to describe the spatial orientation of dentofacial traits analogous to the position of an airplane in space [10] found its way into the analysis of surgical displacements [11,12], with the “barycenter” first mentioned by Schouman and coworkers [13].

For reasons of accuracy, preference should be given to the intraoral scan when planning the postoperative occlusion. The use of plaster models in hybrid planning systems includes various possibilities of error; on the one hand, the lack of dimensional stability and abrasion of plaster [14] and on the other, the difficulty of aligning scanned models to the inaccurate dental surfaces produced by CBCT [7,15].

A new three-dimensional orthognathic surgical planning system based on intraoral scans using arbitrary articulator dimensions has been introduced [16]. It is mandatory to validate any new system to justify the intended benefit for the patient. The aim of this study is therefore to assess the in vivo accuracy of this system in terms of jaw relation and correlation to the extent of surgical movements. In addition, the possible relationship between these and Proffit‘s hierarchy of stability [6] is evaluated. Comparative data from the literature are used to position the new system relative to the accuracy of existing virtual planning systems.

## 2. Materials and Methods

This retrospective cohort study was approved by the local Ethics Commission of the Medical Faculty of the University of Münster, Germany (2021-120-f-S, date of approval: 11 March 2021). According to the guidelines of the Ethics Committee, oral or written patient information and consent are not required for retrospective analysis of intra-departmental data. The study protocol was described according to the STROBE Guidelines [17].

The study took place at the University Hospital Münster, Germany. All patients included in this study were virtually planned with the Digital Münster Model Surgery (DMMS) system [16]. After an interdisciplinary consensus on orthodontic preparation, virtual planning, surgical plan, and intended postoperative occlusion, all patients underwent surgery at the Department of Cranio-Maxillofacial Surgery.

Medical records of all patients planned with the DMMS have been screened from the implementation of digital planning in 2018 to date. Inclusion criteria were completeness of documentation, pre- and postoperative intraoral scans, LeFort I- and bilateral sagittal split osteotomies (BSSO) as one- or two-jaw procedures. Exclusion criteria were all kinds of segmental osteotomies (2-piece-maxilla, Delaire-Joos, Zisser) as well as distractions and LeFort II or III surgeries.

All necessary records for virtual planning were obtained two weeks before surgery. Upper and lower jaws were scanned with TRIOS^®^ Ortho (3Shape, Copenhagen, Denmark), and bite scans were taken after a reproducible mandibular position had been achieved using the Dawson method [18]. The planning sequence is clearly described in a former publication [16]. Surgery was performed with the aid of 3D printed (Solflex 350, W2P Engineering, Vienna, Austria) intermediate and final splints (Ortho Clear, NextDent B.V. Soesterberg, the Netherlands). The segments were fixed using special orthognathic plates (Medartis, MODUS^®^ 2 Orthognatics 1.5/2.0, Basel, Switzerland), which allow transverse flexibility so that the condyle can adjust to the optimal individual position under load during the postoperative functional follow-up. The inpatient stay usually lasted five days and all patients underwent the same postoperative care regimen by the same group of residents. Patients received guiding elastics postoperatively to facilitate adaptation to the new occlusion.

The primary outcome was the spatial difference in the maxillo-mandibulo relation between preoperative virtual planning and postoperative situation measured in terms of translation and rotation of the three spatial axes. The secondary outcome was the possible relation of the measured differences to the extent of the surgical movement and procedures according to the hierarchy of stability in orthognathic surgery [6].

### 2.1. Measurement Process

The postoperative relation was captured by taking new bite scans (Figure 1). Passive orthodontic appliances are a prerequisite for surgery so that dental arches remain stable between planning, splint fabrication, and surgery. Assuming no tooth movement between planning and splint removal, the preoperative stereolithography (STL) files were duplicated and the associated bite scans were deleted. The upper and lower STLs were thus detached and could be reoriented by new postoperative bite scans. The newly oriented STLs were exported for further processing.

To investigate differences between planned and postoperative STL files, a surface-based comparison was performed using GOM Inspect 2018 (Rev. 114010, GOM GmbH, Braunschweig, Germany). The STL files of the upper and lower jaw in planned occlusion were imported as “CAD elements”. The STL files in postoperative occlusion were imported as “actual data elements”. Because there is a greater likelihood that post-operative differences were caused by surgical positioning of the mandible rather than the maxilla [19,20,21], the maxillary record served as the reference for registration. Planned and postoperative jaws were matched through best-fit alignment of the maxillae, which allows the identification of possible deviations between planned and postoperative mandibular positions. Both mandibular data sets were then exported as a polygon file format.

For analyzing the different jaw positions in terms of translation and rotation, domestic software (“pVision3D”) was used [22]. By employing the iterated closest point (ICP) algorithm [23], this software performs a best-fit calculation of the transformation that matches (registers) the two polygon models, predicted mandibular and actual mandibular, thus determining a translation vector and the rotation angles with respect to the three spatial axes.

### 2.2. Statistical Analysis

To calculate the sample size, a preliminary investigation was performed on a test sample. Equivalence tests are provided for this purpose, in which the mean differences are tested against the constant 0. A difference of more than 2 mm was considered clinically relevant, so −2 and 2 are used as equivalence bounds. For each of the five tests, a significance level of α = 0.01 (=0.05/5) and a power of 1 − β = 80% are provided. The calculation resulted in 17 patients being enrolled in the study. Based on the study of Proffit et al. [6] who identified four categories of stability including eight different surgical procedures, it is assumed that the procedures also differ in their planning accuracy. Therefore, it was aimed to include at least 8 × 17 = 136 cases in the study. Nonparametric and one-way ANOVA post hoc Bonferroni tests were used to elicit differences between planning and postoperative outcome. Pearson’s correlation coefficient was calculated to assess a possible relationship between the magnitude of surgical movement and error size. For comparison, the root mean square distances (RMS) commonly used in other studies were also calculated. RMS is the square root of the average of squared movement errors [9] from the transformation matrix of pVision3D. All statistical analyses were performed at a 5% significance level using IBM^®^ SPSS^®^ Statistics 27 (IBM Corp., Armonk, NY, USA).

## 3. Results

Out of 404 eligible cases, 151 patients were finally included in this study. The major reason for exclusion was incomplete records due to missing last check-up appointments. This was because a large proportion of patients were referred for surgery only and postoperative splint care was provided in the hometown.

The study group consists of 90 female (mean age 28.7 ± 8.7 years) and 61 male (mean age 27.7 ± 6.1 years) patients. Female and male patients differed in terms of their skeletal problems, with a significantly higher incidence (*p* ≤ 0.05) of class II in female patients (Table 1).

The average time between preoperative scan and surgery was 21.4 ± 9.2 days and between surgery and postoperative scan 40.3 ± 14.1 days. Sixty-eight patients received single-jaw and eighty-three received two-jaw surgeries. The most used procedures were maxillary advancement combined with posterior impaction followed by mandibular advancement and mandibular clockwise (cw) and counterclockwise (ccw) rotation (Table 2).

### 3.1. Conformity between Planned and Postoperative Positions

The primary outcome measurement was the difference between planned and postoperative maxilla-mandibula relation. The absolute differences for all surgical movements are below the clinically accepted error size of 2 mm, both as a three-dimensional distance in space and in terms of translation along each spatial axis (Table 2). The orientation errors measured around the x-, y-, and z-axes are also below the proposed sufficiently accurate threshold of four degrees [8].

In general, the individual surgical movements do not differ in their error magnitude (ANOVA post hoc Bonferroni, *p* > 0.05). If excluding the one case of single maxillary surgery, the largest three-dimensional distances are found for mandibular setback, maxillary counterclockwise rotation, and all asymmetric movements in the transverse plane (Figure 2). If the individual movements are summarized in spatial planes, similar ratios are seen. The spatial plane as such has no effect on the difference between planned and postoperative outcomes (*p* > 0.05). A slightly different distribution of errors is observed for an orientation around the x-, y-, and z-axis (Figure 3). The smallest errors are seen for rotational movements around the anteroposterior (z-) axis.

### 3.2. Magnitude of Surgical Movement and Error Size

There is no significant relationship between the size of movement and the three-dimensional error size, except for the advancement of the maxilla (Table 2). Although the range of motion is only 5 mm, the three-dimensional error increases with the forward displacement of the maxilla.

### 3.3. Categories of Postoperative Stability and Error Size

A different pattern becomes apparent when the surgical procedures are categorized according to the hierarchy of stability in orthognathic surgery [6]. The smallest three-dimensional error was found for maxilla forward (0.8 ± 0.4 mm, stable) and class II maxilla up with mandible forward (0.7 ± 0.4 mm, fairly stable). These movements also differ significantly (*p* ≤ 0.05) from less stable asymmetric procedures (Figure 4).

Looking at the rotation and translation around and along the individual axes, it could be seen that significant errors exist only for translation in the anteroposterior (z-) direction. Maxilla forward (stable) and class II cases with maxilla up and mandible forward (fairly stable) show the smallest error in the anteroposterior direction (Table 3). Maxilla forward (stable) differs significantly from moderate (*p* ≤ 0.05) and less stable (*p* ≤ 0.05) procedures. Class II cases with maxilla up and mandible forward (fairly stable) differ significantly from highly stable (*p* ≤ 0.05), moderate stable (≤0.05), less stable (≤0.05), and unknown stable (≤0.05) procedures (Table 3). The procedures considered problematic by Proffit are not different from the aforementioned ones.

## 4. Discussion

The most common reason for seeking orthognathic surgery is the correction of occlusion or oral function [24,25,26], followed by facial and psychosocial improvement [27]. With new procedures, it is therefore essential to evaluate the postoperative occlusion, which is the orthodontic basis for achieving a good final intercuspation and the associated masticatory function.

It was shown that using the DMMS [16], the three-dimensional error of the overall process, from preoperative data collection through surgery, to the six-week follow-up, is 0.8 mm ± 0.5 mm and, therefore, below the threshold considered clinically acceptable of two millimeters in translation. The values for pitch (0.9° ± 0.7°), yaw (0.9° ±0.8°), and roll (0.5° ± 0.5°) are also below the clinically accepted four degrees.

Existing studies on the evaluation of virtual planning in orthognathic surgery are characterized by very heterogeneous methods [4,19]. In the majority of studies, 3D radiological data are used, and these are combined with other data such as intraoral scans or scans of plaster models. There is hardly any information on how the postoperative occlusion was set. Manual placement of the plaster casts with subsequent scanning and incorporation into the CT data set still seems to be a standard procedure [28,29,30].

When comparing planned and postoperative 3D positions, registration plays an essential role. Depending on the image acquisition method, surface-based and voxel-based registration are used. Both methods are equal in accuracy, with surface-based registration having the lowest mean deviation for hard tissue but having a larger standard deviation [31]. Registration methods are also used in combination with manually set landmarks. This also concerns the placing of the center of rotation. It is easy to understand that yaw, roll, and pitch produce discrepancies of different sizes depending on where the center of rotation is set. Therefore, manual methods are basically more error-prone than automatic ones.

The registration of a reproducible condyle position is independent of the virtual planning system used. No virtual method, no matter how accurate, can minimize bite registration errors. It is therefore considered as an unsolved difficulty and limits the potential benefit of virtual [13,32] and conventional planning. Which bite registrations are preferred for virtual planning is not known, but the bilateral manipulation by Dawson with a mean accuracy of 0.207 ± 0.02 mm [33] seems to be sufficiently accurate [29].

Despite the enormous heterogeneity in the evaluation methods used, all studies attest to the high accuracy of the virtual planning examined. Three systematic reviews investigated the error size in translation and rotation of existing virtual planning systems. Haas et al. [19] identified nine studies with a total of 137 class II or class III patients. In all studies, the average values for translation were less than 1.2 mm (Figure 5) and less than 1.8 degrees for yaw, roll, and pitch (Figure 6). Alkhayer et al. [4] identified 12 studies that included a total of 332 class II or class III patients. The translation errors found were less than 1.85 mm (Figure 5) and rotation differences were less than 2.75 degrees (Figure 6). A recent systematic review by Todin et al. [34] included 27 studies with a total of 719 patients. Translation and rotation errors were smaller than 2.8 mm and 2.7 degrees, respectively (Figure 5 and Figure 6). Although the methods of the existing studies vary substantially, the comparative values attest to the high accuracy of the process evaluated here. It is the error of the overall process, from data acquisition through surgery to follow-up, that is presented here. Since this process shows very good results clinically, it can be concluded that virtual planning with the DMMS itself also has a low error rate.

Nevertheless, there are differences in the surgical procedures used. The fact that individual movements in certain levels are more prone to error could not be confirmed. The magnitude of the surgical displacement also has no influence on the error, except for maxilla forward. Here, positioning by the means of an intermediate splint over the mandible can be the main reason for errors in forward displacement.

A completely different situation arises when surgical procedures are grouped ac- cording to the hierarchy of stability [6]. Although Proffit’s results refer to the long-term changes 1 to 5 years post-treatment, differences were found after about 6 weeks, which Proffit also identified. So, there is evidence that the different stabilities already have their origin in the planning and the operation, and that it is not necessarily solely the insufficient muscular adaptation over time. The increase in error sizes observed in our collective is consistent with the increasingly less stable procedures published by Proffit. The only exceptions are isolated mandibular advancement and maxillary upward displacement. Class II cases with maxilla up and mandible forward are the most stable procedures and also show the lowest error size in our collective. It has been shown that the significant errors occur mainly in the anteroposterior translation (Figure 4) and, thus, deviates from error-proneness in the vertical plane described in the literature. The reason for this could be that no study has yet analyzed the error size regarding the hierarchy of stability. The counterclockwise rotation of the mandible was not separately analyzed by Proffit. However, due to the size of the error, they seem to belong to fairly to moderate stability. Further research is needed on this.

### Strength and Limitations of the Study

One limitation of this study is that the DMMS can only be evaluated indirectly as one step of many in the process of orthognathic surgery. It is therefore not possible to assess how exactly, for example, the isolated maxilla or mandible has been transferred to its new position. However, it can be assumed that if one jaw is mispositioned, this would naturally also affect the planned occlusion of both jaws.

This limitation also applies to segmental osteotomies (2-piece-maxilla, Delaire-Joos, Zisser) as well as distractions and LeFort II or III surgeries. Measuring the accuracy of those surgical procedures CT or CBCT protocols with a reference to the cranial base are necessary where voxel-based registrations are preferred [35,36,37,38].

Another limitation of our method is the lack of possibility to verify long-term postoperative stability. In our study, however, tooth movement within the perioperative six weeks with passive orthodontic archwires is negligible; postoperative orthodontics precludes comparison with preoperative occlusion and thus long-term assessments. Of course, minor tooth movement may occur during the perioperative period, but this error is eliminated by preoperative control of the splint.

On the one hand, an occlusion-based methodology may be a methodological-technical limitation, but on the other hand, it is a clinical strength, because the decisive factor is the postoperative situation, which must be within the orthodontic treatment possibilities. For example, a precisely operated maxilla alone is no guarantee for a class I intercuspation at the end of the treatment.

In contrast to landmark-based registrations, the semi-automated approach used here using best-fit alignment followed by the iterated closest-point algorithm over pVision3D [20] can be considered a strength of the study. Although the placement of landmarks has been confirmed to have high reproducibility, the error caused by the identification of the landmarks ranged from 0.02 to 2.47 mm [30,35]. Depending on the distance to the teeth, even a slight rotation around the centroid of the jaw can cause a significant bite opening or other malocclusion.

The high number of different surgical procedures included allows an evaluation according to the hierarchy of stability. Accuracy and stability are not directly comparable and therefore the present study cannot make any statement on postoperative stability. Despite this limitation, it seems that certain combinations of procedures not only have an effect on long-term stability, but also on the planning and surgical process.

A further limitation of the study is the limited number of skeletal asymmetries. There are a certain number of asymmetric surgical movements, but the dental and skeletal parameters to describe the asymmetry are missing. Further studies on asymmetry with clear radiological inclusion criteria seem useful.

## 5. Conclusions

The overall process evaluated here, from virtual planning through surgery, to the six-week follow-up, shows an error size below the clinically accepted limit of two millimeters in translation and four degrees in rotation. The measured error values are smaller than those of the previously evaluated planning systems.

The magnitude of surgical movement has no influence on the error size, except for maxilla advancement. The different surgical procedures according to the hierarchy of stability in orthognathic surgery have a significant effect on the error size, especially in the anteroposterior direction.

There is evidence that different postoperative stabilities have their origin already in the planning and surgery and not necessarily only in insufficient muscular adaptation over time.

## Figures and Tables

**Figure 1 jpm-12-00843-f001:**
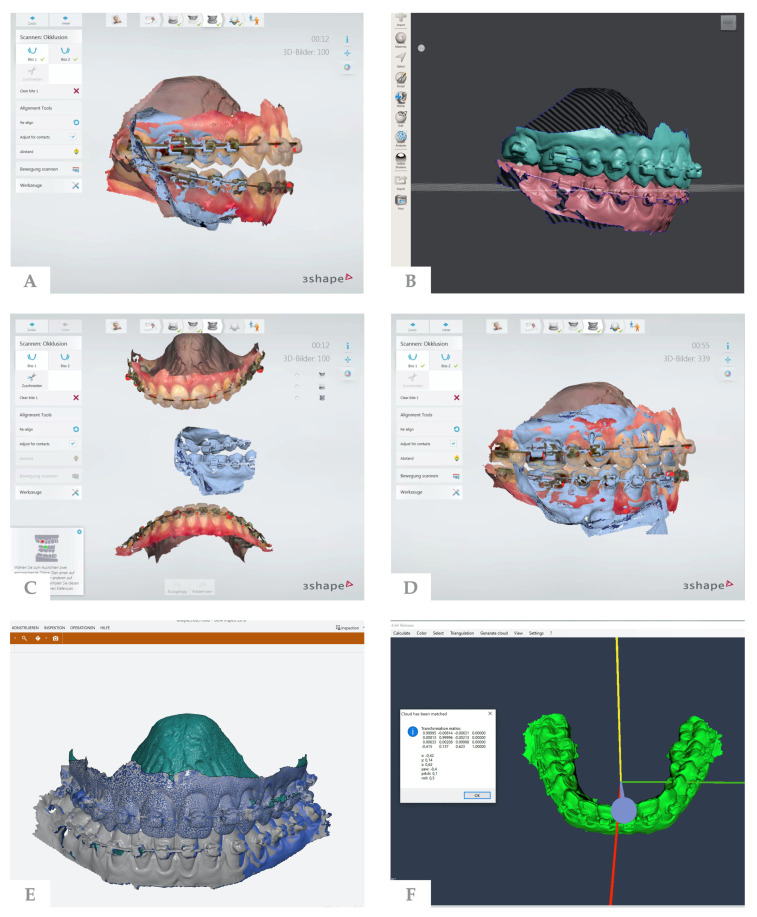
Evaluation process: (**A**) Preoperative upper and lower jaw scans aligned to each other via bite scans. (**B**) Final occlusion after virtual surgical planning of mandibular advancement. (**C**) On the day of the six-week follow-up examination. Based on the passive wires, it is assumed that no tooth movement has occurred between planning and follow-up. To obtain the postoperative occlusion the preoperative scans were duplicated, the bite scans were deleted and, thus, the relation of the upper to the lower jaw was removed. (**D**) New bite scans aligned the jaws in their postoperative position. (**E**) Planned occlusion (CAD element) and postoperative occlusion (actual data elements) were imported with GOM Inspect 2018. The maxilla served as a reference for registration. (**F**) Calculated translation matrix in pVision3D.

**Figure 2 jpm-12-00843-f002:**
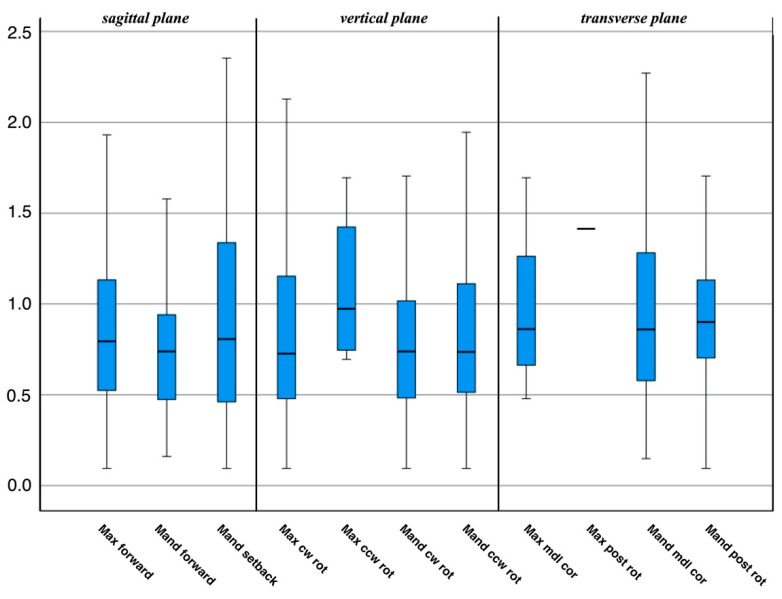
Three-dimensional distances between virtual planning and postoperative outcome categorized according to surgical movements and spatial planes. Less accurate movements are mandible back, maxilla posterior down, and all movements in the transverse plane. Abbreviations: cw, clockwise; rot, rotation; ccw, counterclockwise; mdl, midline; cor, correction; post, posterior.

**Figure 3 jpm-12-00843-f003:**
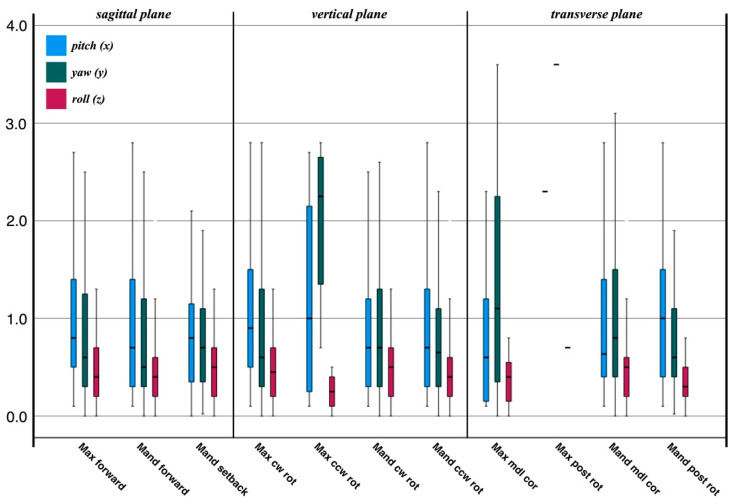
Rotation error around the mediolateral (pitch), superoinferior (yaw), and anteroposterior (roll) axes between virtual planning and postoperative outcome. The smallest errors are seen for rotational movements around the anteroposterior (z-) axis. Abbreviations: cw, clockwise; rot, rotation; ccw, counterclockwise; mdl, midline; cor, correction; post, posterior.

**Figure 4 jpm-12-00843-f004:**
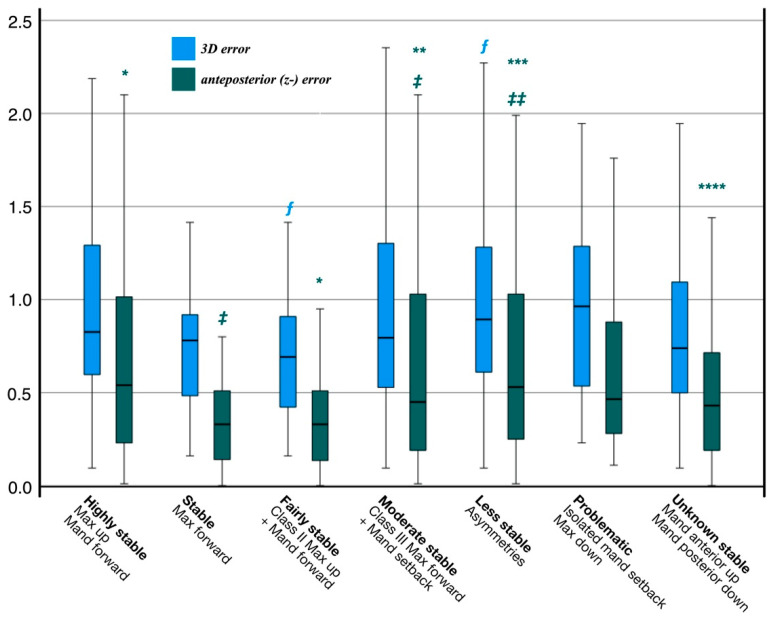
Surgical movements grouped by the hierarchy of stability in orthognathic surgery [6]. There are significant differences in three-dimensional and anteroposterior distances between virtual planning and postoperative outcome (see also Table 3). The lowest errors are seen in stable and fairly stable procedures and differ significantly from moderate, less, and unknown stable procedures. The procedures mandible anterior up and posterior down are not included in the hierarchy of stability and are therefore referred to as unknown stable. * to *; ‡ to ‡; ** to **; *** to ***; ‡‡ to ‡‡, **** to **** indicates significant anteposterior (z-) error between procedures. ƒ to ƒ indicates significant 3D error between procedures.

**Figure 5 jpm-12-00843-f005:**
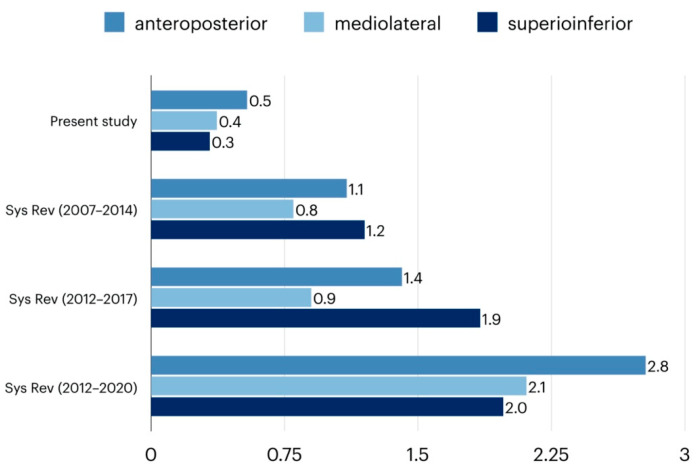
Overview of translational errors concerning studies on virtual planning in orthognathic surgery. Measurements of the present study compared to the results of three systematic reviews (Sys Rev) covering the years 2007–2020 [4,26,35].

**Figure 6 jpm-12-00843-f006:**
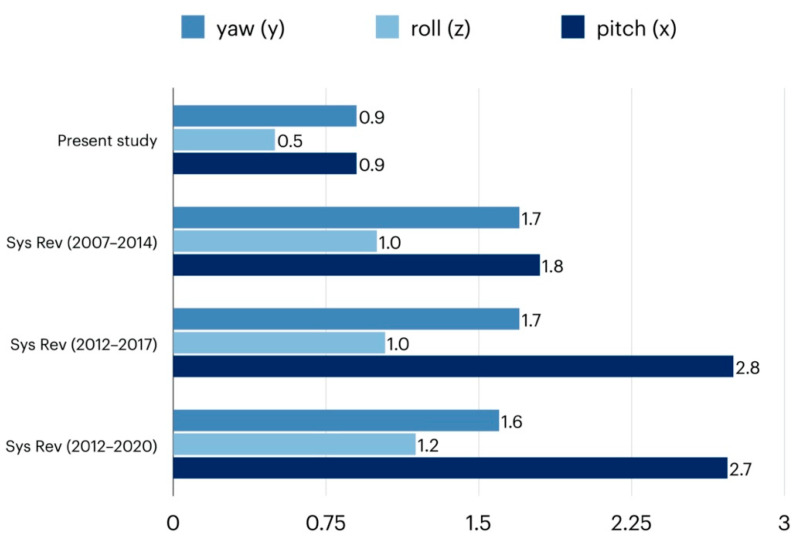
Overview of rotation errors (yaw, roll, and pitch) concerning studies on virtual planning in orthognathic surgery. Measurements of the present study compared to the results of three systematic reviews (Sys Rev) covering the years 2007–2020 [4,26,35].

**Table 1 jpm-12-00843-t001:** Characteristics of the study sample.

Gender	*n*	Age Years	Class II	Class III	Open Bite *
**Female**	90	28.7 ± 8.7	58 (*p* ≤ 0.05) **	32	21
**Male**	61	27.7 ± 6.1	25 (*p* = N.)	36	24

* Number of class II or class III patients with an open bite. ** Nonparametric Mann–Whitney U test. (N. = not significant).

**Table 2 jpm-12-00843-t002:** Surgical procedures and error measurements grouped by spatial planes.

Plane/Procedure	Single-	Two-	Range	3D Error	R	RMS	|x|	|y|	|z|	Pitch (x)	Yaw (y)	Roll (z)
	Jaw	Jaws	(mm)	(mm)		(mm)	(mm)	(mm)	(mm)	(°)	(°)	(°)
**Sagittal plane**												
Max forward	1	82	4 (1–5)	0.9 ± 0.5	0.423, *p* < 0.001	1.0	0.4 ± 0.3	0.3 ± 0.3	0.6 ± 0.6	1.0 ± 0.8	0.9 ± 0.9	0.5 ± 0.5
Mand forward	56	37	10 (1–11)	0.8 ± 0.4	−0.096, *p* = N.	0.9	0.4 ± 0.3	0.4 ± 0.3	0.5 ± 0.4	0.9 ± 0.8	0.9 ± 0.8	0.5 ± 0.5
Mand setback	8	43	8 (1–9)	1.0 ± 0.6	−0.242, *p* = N.	1.1	0.4 ± 0.3	0.3 ± 0.3	0.7 ± 0.7	0.8 ± 0.6	0.8 ± 0.8	0.6 ± 0.5
**Vertical plane ^1^**												
Max cw rotation ^2^	1	69	5 (1–6)	0.9 ± 0.6	−0.085, *p* = N.	1.0	0.4 ± 0.3	0.4 ± 0.3	0.5 ± 0.6	1.0 ± 0.8	1.0 ± 0.9	0.5 ± 0.4
Max ccw rotation	0	4	3 (1–4)	1.1 ± 0.5	0.080, *p* = N.	1.2	0.7 ± 0.3	0.3 ± 0.1	0.7 ± 0.5	1.2 ± 1.2	2.0 ± 0.9	0.3 ± 0.2
Mand cw rotation	29	52	5 (1–6)	0.9 ± 0.5	0.094, *p* = N.	1.0	0.4 ± 0.3	0.4 ± 0.3	0.5 ± 0.5	0.9 ± 0.7	0.9 ± 0.8	0.6 ± 0.5
Mand ccw rotation	42	52	7 (1–8)	0.9 ± 0.5	0.138, *p* = N.	1.0	0.4 ± 0.3	0.3 ± 0.2	0.6 ± 0.5	0.9 ± 0.7	0.8 ± 0.7	0.5 ± 0.4
**Horizontal plane ^1^**												
Max midline correction	1	6	5 (1–6)	1.0 ± 0.5	0.685, *p* = N.	1.1	0.5 ± 0.4	0.3 ± 0.2	0.6 ± 0.5	0.8 ± 0.8	1.4 ± 1.3	0.4 ± 0.3
Max post-rotation	0	1	0 (2–2)	1.4	−	1.4	1.2	0.4	0.7	2.3	3.6	0.7
Mand midline correction	21	37	7 (1–8)	1.0 ± 0.6	0.108, *p* = N.	1.1	0.4 ± 0.3	0.3 ± 0.3	0.7 ± 0.6	0.9 ± 0.7	1.0 ± 0.8	0.6 ± 0.6
Mand post-rotation	6	19	3 (1-4)	1.0 ± 0.5	−0.075, *p* = N.	1.1	0.4 ± 0.4	0.3 + 0.3	0.7 ± 0.5	1.0 ± 0.8	0.7 ± 0.5	0.4 ± 0.4

^1^ Only movements ≥ 2 mm were counted; ^2^ posterior impaction only. (3D error) Mean value ± standard deviation (sd) of the three-dimensional difference between planned and postoperative outcome. (R) Pearson’s correlation coefficient for distance of surgical movement and 3D error. (RMS) Root mean square distances. (x, y, z) Absolute mean ± sd differences according to the spatial axes. (pitch, yaw, roll) Rotation around the spatial axes. (N.) Not significant. (°) degrees.

**Table 3 jpm-12-00843-t003:** Surgical procedures categorized according to Proffit’s hierarchy of postoperative stability in orthognathic surgery. Absolute mean ± sd differences and p-values for significant differences in anteroposterior (z-) direction.

Stability	Procedures	|z| (mm)	*p*
Highly stable	Maxilla up ^1^	0.7 ± 0.6	*
	Mandible forward		
Stable	Maxilla forward	0.4 ± 0.4	*‡* (*p* ≤ 0.05) *‡‡* (*p* ≤ 0.05)
Fairly stable ^2^	Class II (maxilla up + mandible forward)	0.4 ± 0.4	* (*p* ≤ 0.05) ** (*p* < 0.001) *** (*p* ≤ 0.05) **** (*p* ≤ 0.05)
Moderate stable ^2^	Class III (maxilla forward + mandibular setback)	0.7 ± 0.7	*‡* **
Less stable ^2^	Asymmetries	0.7 ± 0.6	*‡‡* ***
Problematic	Isolated mandibular setback	0.6 ± 0.5	
	maxilla down ^3^		
Unknown stable ^4^	Mandibula anterior up	0.6 ± 0.5	****
	Mandibula posterior down		

^1^ In the present study predominantly posterior impaction. ^2^ Proffit summarizes these procedures as stable only with rigid fixation. ^3^ In the present study posterior down. ^4^ Procedures not analyzed by Proffit et al., stability unknown. * to *; ‡ to ‡; ** to **; *** to ***; ‡‡ to ‡‡, **** to **** indicates significant differences between procedures.

## Data Availability

Not applicable.

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
