# Peer review of "In Vivo Accuracy of a New Digital Planning System in Terms of Jaw Relation, Extent of Surgical Movements and the Hierarchy of Stability in Orthognathic Surgery"

_jpm, 2022, doi:10.3390/jpm12050843_

Round 1
Reviewer 1 Report
The manuscript entitled "In Vivo Accuracy of a New Digital Planning System in Terms 2 of Jaw Relation, Extent of Surgical Movements and the Hierar-3 chy of Stability in Orthognathic Surgery. This manuscript is clear, well-write and with the methodology well-conducted. There are few suggestions below:
Introduction
Line 57- I suggested the authors add some information about each rotation: pitch (mediolateral), yaw (superoinferior), and roll (anteroposterior).
Results
The authors must present the p-value "(p =N.)" when there was not significant difference. When the results were significant, put only "(p≤0.05)".It is missing in the tables lower text: 1.The name of statistical test used2. *Statistically significant difference (p≤ 0.05)Add the p-value columns in the tables.When the p-value is < 0.001, put"<0.001"I suggest removing the third sentence of topic “3.2. Magnitude of surgical movement and error size” because the correlations were not significant
Author Response
Dear Reviewer,
Thank you for taking the time to review our manuscript. We are grateful for the overall positive feedback and cherish your constructive suggestions and the valuable input, that will help improve our manuscript. We hope that your concerns and suggestions were addressed appropriately. In the following we give a point-by-point reply to your comments. We have also updated the literature search and the statistical analyses to include recently published articles. You will find changes throughout the body of the manuscript as well as updated figures and tables, also addressing the suggestions from the other referee, as well as a minor English grammar and spelling check by a native speaker. We would like to thank you once again for your willingness to contribute to our research as a reviewer, so that we can improve our study and enhance the accuracy of our digital planning system. We hope that you will consider recommending our paper for publication.
Thank you one more time for your attention.
On behalf of all co-authors
Kind regards,
Eugenia Andriyuk
Reviewer's note: Line 57- I suggested the authors add some information about each rotation: pitch (mediolateral), yaw (superoinferior), and roll (anteroposterior).
Response: Dear Reviewer, we have changed the following sentence:
“Another common feature is that the differences in jaw positions are increasingly analyzed in terms of translation and rotation of the three spatial axes, with rotations described as yaw, roll, and pitch.”
to:
“Another common feature is that the differences in jaw positions are increasingly analyzed in terms of translation and rotation of the three spatial axes, with rotations described as yaw (superoinferior axis), roll (anteroposterior axis), and pitch (mediolateral axis).”
Reviewer's note: The authors must present the p-value "(p =N.)" when there was not significant difference. When the results were significant, put only "(p≤0.05)".
Response: Dear Reviewer, we changed the p-values throughout the text with p=N. for not significant and p≤0.05 for significant values.
Reviewer's note: It is missing in the tables lower text: 1. The name of statistical test used2. *Statistically significant difference (p≤ 0.05) Add the p-value columns in the tables. When the p-value is < 0.001, put" <0.001"
Response: Dear Reviewer, we added the p-values in table 1, as well as a footnote regarding the statistical test used here “**nonparametric Mann-Whitney U test. [N. = not significant]”.
In table 2 we removed column “Max” (please see reviewer 2 note 6) and added the p-values to the R-values in the R-column, e.g. r=0.423, p<0.001 etc. In the footnote we removed “*p < 0.000. [Max] Measured maximum value of the surgical movement.”
Reviewer's note: I suggest removing the third sentence of topic “3.2. Magnitude of surgical movement and error size” because the correlations were not significant.
Response: Dear Reviewer, due to both last sentences indicating not significant correlations and describing the same possible but inexistent relationship, in terms of measurable, statistical significance, we decided to remove both last sentences of 3.2.
We appreciate the time and effort that you and the other reviewer dedicated. We tried our best to meet all the suggestions by the referees and improve the manuscript and hope that the correction will be met with your approval. All co-authors approved the final version of the revision.
Thank you very much.
Yours sincerely,
Eugenia Andriyuk
Reviewer 2 Report
The presented study evaluates the in vivo accuracy of a new planning system. The topic is fitting well to the scope of the journal, as every single orthognathic surgery is personalized, in terms of both preoperative planning, execution and assessment of the surgical outcome of the individual patient. In general, the manuscript is well-written and -organized, and the accuracy of the evaluated DMMS is impressive. Below, several important comments and suggestions are listed, which should be addressed by the authors in a major revision before possible publication.
- By solely relying on pre- and postoperative intraoral scans for deriving the accuracy, the actual amount of postoperative movement cannot be determined for double-jaw procedures. Furthermore, it is not possible to measure long-term postoperative stability of the Jaw segments after passive orthodontic cannot be assumed. This is a limitation of the presented setup and should be discussed in relation to state-of-the-art methods:
(36) Baan F, Sabelis JF, Schreurs R, van de Steeg G, Xi T, van Riet TCT, et al. Validation of the OrthoGnathicAnalyser 2.0—3D accuracy assessment tool for bimaxillary surgery and genioplasty. PLoS ONE. 2021;16(1 January):1–12. doi:10.1371/journal.pone.0246196.
Holte MB, Diaconu A, Ingerslev J, Thorn JJ, Pinholt EM. Virtual analysis of segmental bimaxillary surgery: a validation study. Journal of Oral and Maxillofacial Surgery. 2021 Nov;79(11):2320-33. doi: 10.1016/j.joms.2021.06.003
Holte MB, Diaconu A, Ingerslev J, Thorn JJ, Pinholt EM. Virtual surgical analysis: long-term cone beam computed tomography stability assessment of segmental bimaxillary surgery. Int J Oral Maxillofac Surg. 2022 Mar;26:S0901-5027(22)00070-4. doi: 10.1016/j.ijom.2022.03.007
Andriola FO, Haas Junior OL, Guijarro-Martínez R, Hernández-Alfaro F, Oliveira RB, Pagnoncelli RM, Swennen GR. Computed tomography imaging superimposition protocols to assess outcomes in orthognathic surgery: a systematic review with comprehensive recommendations. Dentomaxillofac Radiol. 2022 Mar 1;51(3):20210340. doi: 10.1259/dmfr.20210340.
- The method fully rely on that the orthodontic is passive from acquisition of the preoperative intraoral scan to the acquisition of postoperative scan. This is assumption may be legit and is argued for based on the use of passive wires. However, it is important to mention that a small movement on dental level can result in a larger deviation on the osteotomy sites.
- It would be interesting to see how the approach performs on segmental osteotomies, especially with regard to what is pinpointed in 1-2. A perspective on this should be added in the discussion.
- In the method section, the authors state: “Because there is a greater likelihood that post-operative differences were caused by surgical positioning of the mandible rather than the maxilla, the maxillary record served as the reference for registration.” A reference is missing to support this claim. Furthermore, by having the maxilla as reference, how do we know if inaccuracies manifest in the maxilla or the mandible? I would like to see a discussion on this issue.
- The authors state: “Sixty-eight patients received single-jaw and 83 two-jaw surgery.” Based on the described method for assessing the accuracy, I would presume that the single jaw procedures all were mandible only surgery. However, in Table 2, I see that single-jaw procedures include forward, CW rotation and midline correction movements of the maxilla. If this is correct, this does not comply with the applied method for measuring the accuracy. Please explain how you measured the accuracy of maxilla only procedures.
- In Table 2, the maximum surgical moment is listed without a measure of variability in the study sample. Is the maximum calculated as the maximum movement of the jaws in all subjects? If so, the number is a poor representation of the study sample. The mean and standard deviation or range would be more representative.
- Please clarify how the RMS distance was calculated – was it between the planned and actual position of the surfaces of the dental models?
- The authors state: “Another limitation, which can also be found in other studies, is the use of the centroid or manually set landmarks for differential measurements.” Which landmarks were used for differential measurements? A description of this is missing in the method section.
- The study presents the accuracy of a new planning system. However, although appearing legit, the validity nor reliability of the method used to measure the accuracy of the planning system is not provided.
- More recent systematic reviews on the accuracy of virtual planning in orthognathic surgery are available:
Tondin GM, Leal MOCD, Costa ST, Grillo R, Jodas CRP, Teixeira RG. Evaluation of the accuracy of virtual planning in bimaxillary orthognathic surgery: Systematic review. Br J Oral Maxillofac Surg. 2021 Sep 20:S0266-4356(21)00343-0. doi: 10.1016/j.bjoms.2021.09.010
Author Response
Dear Reviewer,
Thank you for taking the time to review our manuscript. We are grateful for the overall positive feedback and cherish your constructive suggestions and the valuable input, that will help improve our manuscript. We hope that your concerns and suggestions were addressed appropriately. In the following we give a point-by-point reply to your comments. We have also updated the literature search and the statistical analyses to include recently published articles. You will find changes throughout the body of the manuscript as well as updated figures and tables, also addressing the suggestions from the other referee, as well as a minor English grammar and spelling check by a native speaker. We would like to thank you once again for your willingness to contribute to our research as a reviewer, so that we can improve our study and enhance the accuracy of our digital planning system. We hope that you will consider recommending our paper for publication.
Thank you one more time for your attention.
On behalf of all co-authors
Kind regards,
Eugenia Andriyuk
Reviewer's note 1: By solely relying on pre- and postoperative intraoral scans for deriving the accuracy, the actual amount of postoperative movement cannot be determined for double-jaw procedures.
Response 1: Dear Reviewer, we fully agree with your statement, which is why we were trying to address this issue under 4.2. Strengths and limitations of the study: “One limitation of this study is that the DMMS can only be evaluated indirectly as one step of many in the process of orthognathic surgery. It is therefore not possible to assess how exactly, for example, the isolated maxilla or mandible has been transferred to its new position. However, it can be assumed that if one jaw is mispositioned, this would naturally also affect the planned occlusion of both jaws.”
To address your concern, we also added the following text:
This limitation also applies to segmental osteotomies (2-piece-maxilla, Delaire-Joos, Zisser) as well as distractions and LeFort II or III surgeries. Measuring the accuracy of those surgical procedures CT or CBCT protocols with a reference to the cranial base are necessary where voxel-based registrations are preferred (Baan 2021, Holte 2021, Holte 2022, Andriola 2022).
Furthermore, it is not possible to measure long-term postoperative stability of the jaw segments after passive orthodontic cannot be assumed. This is a limitation of the presented setup and should be discussed in relation to state-of-the-art methods:
We added:
Another limitation of our method is the lack of possibility to verify long-term postoperative stability. In our study, however, tooth movement within the perioperative six weeks with passive orthodontic archwires is negligible, postoperative orthodontics precludes comparison with preoperative occlusion and thus long-term assessments.
Reviewer's note 2: The method fully relies on that the orthodontic is passive from acquisition of the preoperative intraoral scan to the acquisition of postoperative scan. This is assumption may be legit and is argued for based on the use of passive wires. However, it is important to mention that a small movement on dental level can result in a larger deviation on the osteotomy sites.
Response 2: Dear Reviewer, we added to the paragraph above:
“Of course, minor tooth movement may occur during the perioperative period, but this error is eliminated by preoperative control of the splint.”
We would like to add that, rematching the jaw scans with a new bite scan is only possible if the postoperative surface is identical to the preoperative scan. Therefore, this can be considered as an additional validation for the absence of any significant an therefore negligible tooth movement.
Reviewer's note 3: It would be interesting to see how the approach performs on segmental osteotomies, especially regarding what is pinpointed in 1-2. A perspective on this should be added in the discussion.
Response 3: Dear Reviewer, we addressed this limitation under your first note. Segments of the jaws could be planned with the presented system but the used method, which depends on duplicates of the preoperative jaws, is not suitable for segments. Therefore, we excluded segmental osteotomies. To address this issue, we added the text seen under Response 1.
Reviewer's note 4: In the method section, the authors state: “Because there is a greater likelihood that post-operative differences were caused by surgical positioning of the mandible rather than the maxilla, the maxillary record served as the reference for registration.” A reference is missing to support this claim. Furthermore, by having the maxilla as reference, how do we know if inaccuracies manifest in the maxilla or the mandible? I would like to see a discussion on this issue.
Response 4: Dear Reviewer, we discussed this under 4.2. Strength and limitations of the study (please see note 1):
“One limitation of this study is that the DMMS can only be evaluated indirectly as one step of many in the process of orthognathic surgery. It is therefore not possible to assess how exactly, for example, the isolated maxilla or mandible has been transferred to its new position. However, it can be assumed that if one jaw is mispositioned, this would naturally also affect the planned occlusion of both jaws.”
For a discussion of problems related to surgical positioning of the mandible, please see Discussion, 5th paragraph:
“The registration of a reproducible condyle position is independent of the virtual planning system used. No virtual method, no matter how accurate, can minimize bite registration errors. It is therefore considered as an unsolved difficulty and limits the potential benefit of virtual [13] [31] and conventional planning.”
To support this, we added the following references in the method section:
“Because there is a greater likelihood that post-operative differences were caused by surgical positioning of the mandible rather than the maxilla ([26], Lin 2018, Shirota 2019), the maxillary record served as the reference for registration.”
Reviewer's note 5: The author’s state: “Sixty-eight patients received single-jaw and 83 two-jaw surgery.” Based on the described method for assessing the accuracy, I would presume that the single jaw procedures all were mandible only surgery. However, in Table 2, I see that single-jaw procedures include forward, CW rotation and midline correction movements of the maxilla. If this is correct, this does not comply with the applied method for measuring the accuracy. Please explain how you measured the accuracy of maxilla only procedures.
Response 5: Dear Reviewer, thank you for your very accurate observation, which we absolutely confirm. In fact, there is only one isolated maxillary surgery in our patient population and according to your comment, this maxilla was advanced, posteriorly impacted with anterior midline correction. It would have been easy to exclude this case, but it does not fall under the exclusion criteria, so a statement had to be made here as well. As stated under 4.2. Strength and limitations of the study: “It is therefore not possible to assess how exactly, for example, the isolated maxilla or mandible has been transferred to its new position. However, it can be assumed that if one jaw is mispositioned, this would naturally also affect the planned occlusion of both jaws.” In the case of the single upper jaw surgery, this means that although we matched the maxilla and measured an error in the maxillo-mandibulo-relation, we attributed this error to the upper jaw only, since the mandible was not operated on.
Reviewer's note 6: In Table 2, the maximum surgical moment is listed without a measure of variability in the study sample. Is the maximum calculated as the maximum movement of the jaws in all subjects? If so, the number is a poor representation of the study sample. The mean and standard deviation or range would be more representative.
Response 6: Dear Reviewer, we agree that the range would be a better representation. Table 2 was changed accordingly.
Reviewer's note 7: Please clarify how the RMS distance was calculated – was it between the planned and actual position of the surfaces of the dental models?
Response 7: Dear Reviewer, we added to 2.2. Statistical analysis:
“RMS is the square root of the average of squared movement errors [9] from the transformation matrix of pVision3D.”
Reviewer's note 8: The author’s state: “Another limitation, which can also be found in other studies, is the use of the centroid or manually set landmarks for differential measurements.” Which landmarks were used for differential measurements? A description of this is missing in the method section.
Response 8: Dear Reviewer, this sentence has been expressed linguistically inelegantly. What we wanted to point out instead, is that setting landmarks is more error-prone than the semi-automated approach that we used, and which can therefore be considered a strength of the study. To clarify, we therefore added the following under 4.2. Strength and limitations of the study:
“In contrast to landmark-based registrations, the semi-automated approach used here using best-fit alignment followed by iterated closest-point algorithm over pVision3D [20] can be considered a strength of the study. “
Reviewer's note 9: The study presents the accuracy of a new planning system. However, although appearing legit, the validity nor reliability of the method used to measure the accuracy of the planning system is not provided.
Response 9: Dear Reviewer, while your concern is valid, we stongly believe that a separate validation of the method used here is not necessary. As shown in validation studies on planning (e.g. Shaheen et al. 2018), the aim is to determine the inherent inaccuracy of a software, or the magnitude of human error in planning.
In contrast to other planning methods that use special designed software (OrthoGnathicAnalyser, Proplan, Maxilim, IPS Implants, etc.), our method relies on systems (also from engineering) that have been used several times in research due to their accuracy. This concerns TRIOS Ortho, GOM Inspect, Meshmixer and pVision3D. Nevertheless, we have no doubt that the processing of STLs is error-prone and therefore needs to be validated.
Registration and alignment of pre- and postoperative scans is seen as the most important step in predicting the accuracy (Stokbro 2018), where landmark-based methods show an intra- and interindividual error that does not arise with semi-automated methods. Since we do not use a landmark-based method, we do not need to check the reproducibility. The planning itself is of course decided by the practitioner and here there will be differences from one physician to another, but this is not to be checked as an inaccuracy of the method. However, we believe that in such a process, the accuracy of a single step cannot be evaluated accurately at all, because the surgical implementation is still done by the surgeon, so we also stated in the discussion that “It is the error of the overall process, from data acquisition through surgery to follow-up, that is presented here.”
However, existing validation studies do not differ in their aim from our study: “to compare the 3D virtual planning with the actual surgery outcome” (Xia 2007, Shaheen 2018, Marlière 2019, Maniskas 2021).
Maniskas S, Pourtaheri N, Chandler L, Lu X, Bruckman KC, Steinbacher DM. Conformity of the Virtual Surgical Plan to the Actual Result Comparing Five Craniofacial Procedure Types. Plast Reconstr Surg. 2021;915–24.
Kasper Stokbro, Torben Thygesen, A 3-Dimensional Approach for Analysis in Orthognathic Surgery—Using Free Software for Voxel-Based Alignment and Semiautomatic Measurement,
Journal of Oral and Maxillofacial Surgery, Volume 76, Issue 6, 2018, Pages 1316-1326
Marlière DAA, Demétrio MS, Schmitt ARM, Lovisi CB, Asprino L, Chaves-Netto HDDM. Accuracy between virtual surgical planning and actual outcomes in orthognathic surgery by iterative closest point algorithm and color maps: A retrospective cohort study. Med Oral Patol Oral y Cir Bucal. 2019;24:e243–53.
Baan F, de Waard O, Bruggink R, Xi T, Ongkosuwito EM, Maal TJJ. Virtual setup in orthodontics: planning and evaluation. Clin Oral Investig. 2020;24:2385–93.
Xia JJ, Gateno J, Teichgraeber JF, Christensen AM, Lasky RE, Lemoine JJ, et al. Accuracy of the Computer-Aided Surgical Simulation (CASS) System in the Treatment of Patients With Complex Craniomaxillofacial Deformity: A Pilot Study. J Oral Maxillofac Surg. 2007;65:248–54.
Shaheen E, Shujaat S, Saeed T, Jacobs R, Politis C. Three-dimensional planning accuracy and follow-up protocol in orthognathic surgery: a validation study. Int J Oral Maxillofac Surg [Internet]. International Association of Oral and Maxillofacial Surgery; 2019;48:71–6. Available from: https://doi.org/10.1016/j.ijom.2018.07.011
Reviewer's note 10: More recent systematic reviews on the accuracy of virtual planning in orthognathic surgery are available:
Tondin GM, Leal MOCD, Costa ST, Grillo R, Jodas CRP, Teixeira RG. Evaluation of the accuracy of virtual planning in bimaxillary orthognathic surgery: Systematic review. Br J Oral Maxillofac Surg. 2021 Sep 20:S0266-4356(21)00343-0. doi: 10.1016/j.bjoms.2021.09.010
Response 10: Dear Reviewer, thank you for this up-to-date reference, which was not available to us when we first drafted the manuscript. We have adapted the references and included the systematic review in the discussion. Figures 5 and 6 have been updated accordingly.
We appreciate the time and effort that you and the other reviewer dedicated. We tried our best to meet all the suggestions by the referees and improve the manuscript and hope that the correction will be met with approval. All co-authors approved the final version of the revision.
Thank you very much.
Yours sincerely,
Eugenia Andriyuk
Round 2
Reviewer 2 Report
Dear authors,
Thank you for the thorough revision of your manuscript and your comprehensive responses to my comments. The revision meets and discusses the pinpointed issues properly. Hence, I recommend publication.